# Immunotherapy and Liver Transplantation: A Narrative Review of Basic and Clinical Data

**DOI:** 10.3390/cancers15184574

**Published:** 2023-09-15

**Authors:** Charles-Henri Wassmer, Sofia El Hajji, Xenofon Papazarkadas, Philippe Compagnon, Parissa Tabrizian, Stéphanie Lacotte, Christian Toso

**Affiliations:** 1Division of Abdominal Surgery, Department of Surgery, Faculty of Medicine, Geneva University Hospitals, 1205 Geneva, Switzerland; sofia.elhaijji@hcuge.ch (S.E.H.); xenofon.papazarkadas@hcuge.ch (X.P.); stephanie.lacotte@hcuge.ch (S.L.); christian.toso@hcuge.ch (C.T.); 2Division of Transplantation, Department of Surgery, Faculty of Medicine, Geneva University Hospitals, 1205 Geneva, Switzerland; philippe.compagnon@hcuge.ch; 3Mount Sinai Liver Cancer Program, Tisch Cancer Institute, Icahn School of Medicine at Mount Sinai, New York, NY 10019, USA; parissa.tabrizian@mountsinai.org

**Keywords:** immune checkpoint inhibitors, hepatocellular carcinoma, liver transplantation, acute rejection

## Abstract

**Simple Summary:**

Hepatocellular carcinoma (HCC) represents one of the main indications for liver transplantation. Over recent years, immune checkpoint inhibitor (ICI) therapy has improved its management, making patients with more advanced HCC potential candidates for transplantation. However, acute rejection has been observed after ICI therapy, challenging its safety in transplant settings. We summarize and discuss the preclinical and clinical data exploring the use of ICI prior to and after liver transplantation. We identify a three-month ideal minimum period between ICI and transplantation to decrease the risk of rejection. We also warn about its use after liver transplantation and speak about the need for more robust prospective data in the field.

**Abstract:**

Immune checkpoint inhibitors (ICIs) have improved the management of patients with intermediate- and advanced-stage HCC, even making some of them potential candidates for liver transplantation. However, acute rejection has been observed after ICI therapy, challenging its safety in transplant settings. We summarize the key basic impact of immune checkpoints on HCC and liver transplantation. We analyze the available case reports and case series on the use of ICI therapy prior to and after liver transplantation. A three-month washout period is desirable between ICI therapy and liver transplantation to reduce the risk of acute rejection. Whenever possible, ICIs should be avoided after liver transplantation, and especially so early after a transplant. Globally, more robust prospective data in the field are required.

## 1. Introduction

Hepatocellular carcinoma (HCC) arises in the setting of chronic liver disease [1]. Following the trend of obesity and metabolic syndrome, the incidence of HCC is rising. With 830,000 deaths annually and a global 5-year survival rate of approximately 18%, it remains a major global health issue [2]. Therapeutic strategies for HCC rely on the Barcelona Clinic Liver Cancer staging system [3]. In patients with early-stage HCC (Stage 0 and A), the removal/destruction of a tumor can be achieved by local ablation methods, such as radiofrequency (RFA) and microwave ablation (MWA), surgical resection, and liver transplantation (LT). In the presence of intermediate-stage HCC (stage B), intra-arterial therapies, such as transarterial chemoembolization (TACE) and selective internal radiotherapy (SIRT), are first-choice treatments. In patients with more advanced forms of HCC, systemic treatment can be considered [4].

Since 2007, kinase inhibitors, like sorafenib, have been the first line of systemic treatment, offering a median overall survival (OS) of 11 to 14 months [5]. By 2017, one could see progress with the introduction of immune checkpoint inhibitors (ICIs), which target the programmed death 1 receptor (PD-1) and its ligand (PD-L1). Nivolumab and pembrolizumab, two monotherapy anti-PD-1 monoclonal antibodies, initially failed to show improved survival compared to sorafenib in their respective randomized controlled trials [6,7]. However, more recently, the combination of atezolizumab (anti-PD-L1) and bevacizumab (anti-VEGF) demonstrated better outcomes than sorafenib, with an overall survival of 19 months in patients with advanced HCC [8]. This combination has now become the standard of care for first-line systemic HCC treatment [4,9]. The Himalaya trial also showed a significantly better OS (16.4 months) with the combination of tremelimumab and durvalumab in comparison to sorafenib (13.8 months) [10].

Liver transplantation is considered in patients with single nonresectable or multiple HCCs [11]. It leads to a 5-year survival of approximately 80% when the Milan criteria are respected (one lesion < 5 cm, up to three lesions each being < 3 cm, and the absence of extrahepatic lesions or macrovascular invasion) [12]. Over the years, in order to offer access to transplantation to more patients, extended criteria have been introduced, such as the prospectively validated University of California San Francisco criteria (one lesion ≤ 6.5 cm, up to three lesions with the largest being ≤4.5 cm, and a total tumor diameter of ≤8 cm) [13]. Additionally, one moved from purely morphometric criteria to criteria combining morphology and biology, including the French AFP model (combining AFP, tumor size, and number) and the total tumor volume (≤115 cm^3^)/alpha fetoprotein (≤400 ng/mL) score [14,15].

When patients outside of these criteria have been successfully downstaged prior to being listed, they can also be considered for transplantation [16,17]. However, due to their immunological impact and the potential risk of post-transplant rejection, such situations can be highly challenging when ICIs have been used for downstaging. With the wider adjuvant use of atezolizumab/bevacizumab after liver resection, based on its described improved recurrence-free survival in the IMbrave 050 trial, one may observe an increase in the number of such transplant candidates in the coming years, which is another cause for concern [18].

Studies report acute rejection, graft loss, and death in patients that receive an anti-PD1 antibody (nivolumab, toripalimab, camrelizumab, or pembrolizumab) prior to transplantation, regardless of the time between the end of treatment and transplantation [19,20,21,22,23]. In the absence of guidelines on the topic, these reports raise the question of the safety of transplantation after ICI therapy. Additionally, one still needs to explore the potential underlying mechanism, the need to respect a minimum safety period between the end of ICI therapy and transplantation, and the potential to identify patients at risk of rejection based on immune markers. Finally, one must also assess the adoption of preconditioning measures to prevent rejection. This review aims to summarize the available data on liver rejection in patients that have received ICI treatment before or after liver transplantation, to report on the availability of preclinical data, and to identify areas of future research.

## 2. Preclinical Mechanisms of Rejection

Immunosuppressive drugs and ICI therapy are two opposing immune treatments. The former mitigates the allogeneic response, whereas the latter suppresses the negative immune feedback mechanisms that destroy cancer cells. They share common immune targets, including the programmed death 1 receptor (PD-1) and its ligand, PD-L1, and the cytotoxic T-lymphocyte-associated protein 4 (CTLA-4).

It is well established that, among other aspects, the PD-1/PD-L1 interaction participates in the induction and maintenance of solid organ tolerance. PD-1 and PD-L1 expression is upregulated on activated lymphocytes, and on antigen-presenting cells (APCs) such as dendritic cells, B cells, and monocytes [24,25]. PD-1/PD-L1 interaction results in regulatory T cell (Treg) development and maintenance, suppresses T cell activation and induces their exhaustion [26], and reduces IFN-γ expression [27], which together create a tolerogenic environment that promotes graft survival [28]. As an illustration, PD-1 -/- or PD-L1 -/- recipient mice reject cardiac allograft transplantation, in contrast to control wild-type (WT) mice, even when given immunosuppressive treatment [29]. While grafts survive > 120 days in WT mice using CD154 mAb in association with either donor splenocyte transfusion or CTLA-4Ig, acute rejection is observed in PD-1 -/- or PD-L1 -/- mice, even after the administration of the aforementioned immunosuppressive regimen [29]. Moreover, the benefits obtained in WT mice in graft survival using CD154 mAb + CTLA-4Ig are abrogated when mice are given anti-PD-1 mAb or anti-PD-L1 after transplantation [29,30]. Interestingly, PD-L1-deficient heart grafts are accepted by WT recipients treated with CTLA-4Ig but develop signs of severe chronic rejection and vasculopathy [30]. Similarly, blocking the PD-1/PD-L1 pathway via anti-PD-L1 antibodies or using PD-L1 KO mice as donors led to graft rejection in a mouse model of liver transplantation [31]. These experiments suggest that the expression of PD-L1 on the recipient’s cells (like host antigen-presenting cells) and on graft cells is critical for the maintenance of graft acceptance.

Compared to PD-1/PD-L1, CTLA-4 appears to have less of an impact on allograft acceptance. It participates in the development of an immunoprivileged environment that helps graft acceptance, although it appears to play a less significant role in maintaining it [32,33]. To illustrate this, the use of anti-CTLA-4 early after mice cardiac allograft transplantation leads to rejection, whereas blocking CTLA-4 later after transplantation does not result in rejection [32,34]. Additionally, using a mouse model of graft versus host disease (GVHD), anti-CTLA4-treated animals experienced less GVHD and a lower mortality rate than mice treated with anti-PD-1 [35]. Clinical observations that show that fewer patients appear to experience organ rejection after anti-CTLA-4 treatment compared to anti-PD-1 could support these findings [36,37,38]. Altogether, these data point to a preclinical link between ICIs and allogeneic rejection. Based on the weak data currently available, anti-CTLA-4 may be less of a problem, especially when used late after transplantation.

The underlying liver disease also appears to play a role in the response to ICIs. Using a meta-analysis (CheckMate-459, IMbrave 150, and KEYNOTE-240), Pfister et al. demonstrated that patients with virus-related HCC benefit more from ICI therapy than those with nonviral liver diseases (NASH and NAFLD) [39]. Nonviral liver damages could alter the liver microenvironment and impair the immune surveillance. Patient stratification based on the cause of the liver disease may be needed for personalized HCC therapy.

We summarized the ICI impact on the immune system in pre- and post-transplantation in Figure 1.

## 3. Tregs, ICIs, and Transplantation

Beyond their contribution to various autoimmune diseases [40], Treg cells act in the tumor microenvironment and can promote cancer immune evasion through checkpoint inhibitors [41]. They can reduce the function and expansion of CD4 and CD8 T cells, B cells, natural killer cells, and APCs, and their presence in tumor tissue is associated with poorer oncological outcomes [42]. Interestingly, the increase in immune response secondary to anti-CTLA4 treatment appears more linked to Treg activity than to effector T cells [43]. The Treg depletion and blockage that ICIs induce result in a robust antitumor response [41]. However, this positive effect is sometimes obtained at the cost of autoimmune phenomena, such as hypophysitis with anti-CTLA-4 and hypothyroidism with anti-PD-1 [44]. Globally, Tregs are crucial for allograft tolerance and immunological escape from malignancy. Consequently, Tregs can significantly impact the use of ICIs [45].

## 4. Tissue-Resident Memory T Cells, Exhausted T Cells, ICIs, and Transplantation

Intratumoral subtypes of T cells, their state (naive, effector, dysfunctional, or exhausted), and their contribution to immune homeostasis in the tumor microenvironment have been previously reviewed [46]. Interestingly, the level of T cell dysfunction within a tumor in lung and skin cancer, and the level of protein expression, such as PD-1 and CTLA-4, are predictors of the response to ICI therapy [47,48]. Two categories of T cells have received attention in recent years for their role in cancer and the response to ICIs: tissue-resident memory T (T_RM_) cells and exhausted T (T_EX_) cells.

The presence of T_RM_ (CD103^+^CD69^+^CD8^+^) cells with high expression of PD-1 in the HCC microenvironment is associated with a better response to ICIs and better oncological outcomes [49,50,51]. Adding to these data, a correlation was observed between the quantity of tumor T_RM_ cells and patient survival in melanoma patients [52]. Interestingly, this cell subset showed increased immune checkpoint expression, especially PD-1 and 2B4, which suggests a role in antitumor immunity after ICI treatment. More recently, Barsch et al. went a step further in HCC; they reported that high levels of T_EX_ cells with the increased expression of PD-1 and other immune checkpoints, including LAG-3 and CTLA-4, had a negative impact on the prognoses of patients [53]. They went on to demonstrate that a higher ratio of CD103 + T_RM/_T_EX_ cells within the tumor offers a better prognosis associated with a better response to anti-PD-1 therapy. These findings reveal a high degree of heterogeneity in T cell subpopulations and in various states within the tumor, which is indicative of complex interactions. This may partially account for the conflicting results observed in the literature, such as the absence of a correlation between total tumor-infiltrating CD8 T cells and the response to nivolumab observed in the CheckMate 040 study [54]. Overall, we can assume that intratumoral T_RM_ cells are important surrogates that allow us to predict the response to ICI therapy and, thus, the prognosis. The proportion of T_EX_ cells present in a tumor seems to correlate with poor outcomes.

The role of memory T cells has also been explored after allogenic transplantation [55,56]. Some have suggested that, after repeated exposure to alloantigens, T cells become exhausted over time, which contributes to graft acceptance [57]. However, others have demonstrated that T_RM_ cells migrate to the graft after allogenic mouse kidney or islet transplantation and participate in chronic rejection, and that only a small percentage of them exhibit exhaustion [58,59,60]. Interestingly, a brief course of cyclosporine delayed allogeneic kidney rejection but did not prevent the migration of CD103_+_ T_RM_ cells in the graft [61]. The initial hypothesis behind T_RM_ cell exhaustion and ICI therapy was that ICI therapy targets exhausted T cells, which become reinvigorated and act against tumor cells or, in the case of allografts, participate in rejection. The hypothesis of T cell exhaustion is counteracted by the potential absence of T_RM_ cell exhaustion and may depend on a recipient’s immunosuppressive status. This is reinforced by the absence of rejection even after the repeated administration of anti-PD-1 after transplantation.

However, the link between memory T cells, ICIs, and rejection deserves further exploration, especially in the liver, which allows for a specific tolerogenic environment despite the infiltration of T_RM_ cells [62]. As potential targets for therapies that prevent ICI-induced rejection, they may serve as biomarkers of an increased risk of rejection.

We summarized the implicated immune cells and their actions on HCC, ICI and transplantation in Table 1.

## 5. PD-1/PD-L1 Expression on Tumor, Immune, and Transplanted Liver Cells

The level of PD-L1 expression on tumor cells correlates with the response to ICI therapy in various types of cancers, including non-small-cell lung cancer [63,64]. In the KEYNOTE-224 trial, the combined expression of PD-L1 on immune and HCC cells showed the best prediction for HCC [65]. Furthermore, Kim et al. demonstrated that a subpopulation of CD8 + T cells in the tumor microenvironment that express PD-1 highly can predict tumor aggressiveness and ICI responses [66]. Finally, PD-1 expression could be evaluated in peripheral blood, removing the need for invasive biopsies. Another group also validated the correlation between PD-1 expression in peripheral blood and that within a tumor [67].

Liver grafts do not show systematic PD-L1 expression prior to transplantation [19]. After transplantation, hepatocytes, cholangiocytes, and sinusoids express some level of PDL-1. In parallel, PD-1 was also highly expressed by recipient T cells that had been infiltrated. This combination contributes to the counter-regulation of rejection events, as illustrated by higher rates of rejection associated with specific donor PD-L1 and recipient PD-1 single-nucleotide polymorphisms [68]. Blocking this interaction via ICI therapy also alters the immune-protective state associated with it and can result in acute rejection. Further research should assess the relationship between PD-L1 liver grafts and PD-1 levels of expression and polymorphisms in peripheral T cells, as well as the risk of ICI-promoted rejection.

## 6. Use of ICIs Prior to Liver Transplantation

Immune checkpoint inhibitors can be used as downstaging or bridging therapies prior to liver transplantation for HCC. In patients with HCC, they have shown promising results, with a significantly higher overall survival rate and a longer median progression-free survival [8]. Despite being acceptable in patients outside the transplantation field, their safety in patients on a liver transplant waiting list needs further validation [69,70]. Furthermore, locoregional therapies (LRTs), such as TACE and TARE, are already being widely used in order to reduce and control HCC lesions during the waiting time prior to transplantation. Llovet et al., among others, reviewed the potential to combine LRTs with ICI therapy, aiming to potentialize the effect on tumor control and destruction. After LRTs, necrotic tumor cells release antigens that can be recognized by the increased immune system, boosted by ICI therapies. This will probably shape the future of HCC management in intermediate and advanced HCC and will probably increase the amount of patients receiving ICI therapies on the waiting list [71].

Up to May 2023, we found 14 studies that assessed patients with pretransplant ICIs. These studies included seven case reports [19,20,22,23,72,73,74], six case series [21,75,76,77,78,79], and one multicenter study [80]. With the exception of the study by Tabrizian et al., which included patients prospectively, all reports are single-center, retrospective studies. Study characteristics are presented in Table 2. One review exists and includes six of the aforementioned studies [81]. In total, 54 patients received ICIs prior to liver transplantation, 37 of whom were men, 6 of whom were women, and 11 of whom were of an unspecified gender. The most common underlying liver disease was viral hepatitis, and the most commonly used ICI was nivolumab. Other agents included toripalimab, durvalumab, pembrolizumab, sintilimab, and camrelizumab. The duration of treatment ranged from 6 weeks to 34 months. In total, 20 patients (37.0%) experienced acute rejection and, among them, three patients (5.6%) died because of graft loss. Two patients (3.7%) underwent successful retransplantation and thirteen patients (24.1%) underwent successful treatment through adaptation of the immunosuppressive regimen, corticosteroid use, or antithymocyte globulin treatment. Data are missing for two patients regarding the management of the rejection. Of note, one rejection was attributed to insufficient immunosuppression. Most rejections occurred after nivolumab, except for two patients who received toripalimab and pembrolizumab. Interestingly, nivolumab has a reported half-life of approximately 25 days [82,83]. For the eleven patients with acute rejection, the washout period (the time between the last dose of ICIs and transplantation) ranged from 8 days to 93 days, with most of the patients receiving transplants between 30 and 40 days following the last dose of ICIs. In the cases reported by Dave et al., the washout period for all patients with acute rejection was fewer than 90 days, whereas the washout period for all patients without rejection was longer than 90 days [77]. In the most recent case series, Wang et al. revealed a significant difference in the washout times between the rejection group and the no rejection group, of 21 days (15.5–27.5) and 60 days (24–167), respectively [78]. All patients without rejection had a washout period of at least 24 days. Similarly, in the largest prospective study currently available, Tabrizian et al. found that patients with a washout period longer than 90 days had a significantly lower probability of rejection [80]. They looked at 80 HCC patients who received ICI therapy and were eligible for LT. Sixty-seven percent were downstaged, mainly due to locoregional therapy. In total, 30 patients (37.5%) were transplanted; 33 discontinued treatment (17 due to tumor progression); and 15 were still on the waiting list. A total of five patients (16.6%) experienced allograft rejection, three of whom did so because of low immunosuppression. They found that a shorter washout period—less than 3 months—was linked to a greater rejection rate.

Therefore, it appears that a minimum washout period of 30 days should be respected, if possible, to reduce the risk of rejection. However, even with prolonged washout periods, outliers with rejection have been reported, which suggests that the target occupancy and action of anti-PD-1/PD-L1 exceed their half-lives. After a single administration of nivolumab, Brahmer et al. reported PD-1 occupancy on lymphocytes of up to 100 days [84]. To unravel the underlying mechanisms, preclinical and translational studies are necessary.

## 7. ICIs after Liver Transplantation

The population of liver transplant recipients in need of cancer treatment is growing [85]. This is linked to an increased propensity for cancer recurrence due to the use of extended inclusion criteria combined with the effect of immunosuppression [86,87]. ICI-based treatment for this specific population requires increased investigation, especially regarding the safety, indications, dosages, and durations of treatment [88]. As summarized in Table 3, we found 33 studies that report using ICIs after liver transplantation [89,90,91,92,93,94,95,96,97,98,99,100,101,102,103,104,105,106,107,108,109,110,111,112,113,114,115,116,117,118,119,120]. Altogether, these studies report 57 patients, with a mean age of 57.4 years and the majority being men (75.5%). HCC and decompensated cirrhosis were the main indications for LT. The majority of immunosuppressive regimens were a combination of mycophenolate mofetil and calcineurin/mTOR inhibitors; however, around 26% of patients were under monotherapy, mainly calcineurin inhibitors. Following LT, indications for ICIs were primarily HCC recurrence and novel appearances of melanoma and lung cancers. More than 88% of the patients received anti-PD1 therapy, with nivolumab being the most frequently used. Four patients received a combination of atezolizumab and bevacizumab after LT in three studies without experiencing graft rejection [89,90,91]. The follow-up, however, was short and ranged from 7 to 10 months. With a mortality rate of 12.1%, the liver rejection rate was 25.9%.

Globally, the timing of immunotherapy after transplant appears of most importance. Patients with the longest time after transplantation appear to be less at risk of rejection [121]. The mean time between LT and ICI use was 6 years. Interestingly, Kayali et al. observed that patients responding to ICI therapy had a longer interval between LT and ICI therapy than nonresponder patients (6 vs. 3 years) [122]. Furthermore, though not statistically significant, they observed that patients who experienced graft rejection had a shorter period from LT to ICI therapy (2 vs. 4 years). Additionally, the nonresponder patients showed increased graft rejection. When further in time from LT, the better response to ICI therapy can be partially explained by the progressive reduction in immunotherapy.

In addition to timing, a high PD-1/PD-L1 expression level in the liver graft also appears to be a predictor of rejection. Zhang et al. showed that recipients with positive PD-L1 staining showed increased rejection rates and higher mortality compared to those with no detectable PD-L1 expression [123]. Similarly, Munker et al. evaluated three biopsies with the same findings [124]. The use of a biopsy prior to treatment initiation is supported by the link between positive PD-L1 staining on histology and graft rejection.

## 8. Prevention and Management of ICI-Induced Liver Graft Rejection

In the studies mentioned here, the reported immunosuppression in patients who receive ICI therapy before LT is relatively standard. Most often, it consists of induction with methylprednisolone followed by mycophenolate mofetil (MMF), a calcineurin inhibitor, or occasionally an mTOR inhibitor and prednisone weaned over a few weeks. During induction, some groups also administered basiliximab or antithymocyte globuline (ATG), though this did not completely prevent rejection [21,77,78].

After LT, acute rejection management is relatively standardized, with methylprednisolone used as the first-line treatment and resulting in a 90% success rate in reversing rejection [125]. In cases of steroid-resistant acute rejection, the use of ATG proved to be effective [126,127,128]. The use of plasmapheresis is thought to increase the likelihood of overcoming acute rejection, in part because it may remove ICIs from the organism [129]. In addition, changing baseline immunosuppression may also improve outcomes. However, salvage was not uniform, with a few patients needing retransplantation [23,76].

## 9. Adjuvant ICIs after Curative HCC Treatment

The adjuvant use of ICIs following ablation or surgical resection has been assessed previously [130]. These investigations have been conducted based on the finding that even small HCCs (≤2 cm) have a 10% likelihood of developing intrahepatic metastasis [131]. Among the five ongoing clinical trials, three (NIVOLVE, CheckMate, and KEYNOTE-937) have explored the adjuvant effect of ICIs in monotherapy, using nivolumab for the first two and pembrolizumab for the third. Two other clinical trials explored the benefit of the durvarumab/bevacizumab combination (EMERALD-2) and that of the atezolizumab/bevacizumab combination (IMbrave 050). The latter is a phase 3 randomized clinical study that compares the combination of atezolizumab/bevacizumab against active surveillance in patients at a high risk of recurrence following HCC ablation or resection, including those with an HCC size >5 cm, >three tumors, microvascular invasion, minor macrovascular invasion (Vp1/Vp2), or a grade 3/4 pathology [18,132]. For the first time, the preliminary results of the IMbrave study have demonstrated a statistically significant improvement in recurrence-free survival (RFS) in patients with adjuvant atezolizumab/bevacizumab compared to patients with active surveillance [18]. At 12 months, 78% of patients in the atezolizumab/bevacizumab group had RFS, compared to 65% of those in the active surveillance group (HR = 0.72 (95% CI: 0.56, 0.93; *p* value = 0.012)) [133]. At the first clinical cutoff of 17.4 months, 40% of patients in the active group experienced disease recurrence or death, compared to 33% of those in the ICI group [133]. The 12-month recurrence-free rate in the ICI group was 34%, compared to 20% in the active surveillance group (HR = 0.67 (95% CI: 0.52, 0.88; *p* value 0.003)). Such data could change clinical practice and have a significant impact on the number of ICI patients that are downstaged and ultimately fulfill the criteria for transplantation. Among the dozen of ongoing clinical trials studying the use of atezolizumab/bevacizumab for advanced stages, we found four trials (NCT05185505, NCT04721132, NCT05137899, and NCT05908786) evaluating the benefit of using ICIs in a neoadjuvant setting before liver surgery and only one of them is evaluating its effect before LT (NCT05185505). Most of these trials take place in North America and their results could also impact the number of patients with a history of ICIs use on the waiting lists, especially the trial looking at the possibility to downstage patients outside the Milan Criteria, with 6 months of atezolizumab/bevacizumab, which could allow one to bridge these patients to LT. The ongoing trials are listed in Table 4 (www.clinicaltrials.gov, accessed on 11 September 2023).

This development further supports the need for studies defining ICI management prior to transplantation.

## 10. Conclusions

The field of HCC management is undergoing rapid change, with the most recent advances linked to the introduction of immunotherapy as both primary and adjuvant lines of treatment. They are associated with improved response rates and have the potential to lead patients, via downstaging, toward resection or transplantation. These positive aspects do, however, also present new immune challenges. ICIs are designed to promote immunity that acts against both the HCC and allogeneic liver. Their effect against a graft could be mediated via memory cells and Treg cells and appears more active when PD-1 and PDL-1 are expressed at high levels in the liver.

ICI-promoted rejections have been reported in patients who were responding well to ICIs, have been downstaged, and are now meeting transplant criteria. The most recent case series, however, did not reveal the original, alarmingly high levels of rejection rates described in case reports, most likely due to high reporting bias in the first patients. While further data are needed, transplanting such patients appears feasible. However, a 90-day break from ICIs is desirable prior to a transplant; appropriate immunosuppression should be used, including the use of induction and steroids; and early liver graft biopsy as well as treatment are recommended in the event of a suspected rejection.

The use of ICIs following liver transplantation is more challenging and should only be considered in extremely select and unusual patients, where no alternative oncological option is available. Ultimately, the benefits of ICIs should outweigh any potential harmful effects of rejection, and salvage immunosuppression should be introduced early in the case of a suspected rejection.

## 11. Future Directions

Within the coming years, HCC management will likely further evolve. ICIs could be used in combination with locoregional treatments, and as neoadjuvants/downstaging in view of surgery or transplantation. As a result, the number of patients exposed to ICI therapy on the waiting list could increase. We collectively need to explore a number of key questions: Based on their higher risk of post-transplant recurrence, which patients would benefit from pretransplant ICIs the most? What should this selection be based on? Additionally, which ICI drug should be favored? What is the best immunosuppression strategy? Is liver donor liver transplantation acceptable? Such data will allow for the safer use of peritransplant ICIs and globally for HCC management.

## Figures and Tables

**Figure 1 cancers-15-04574-f001:**
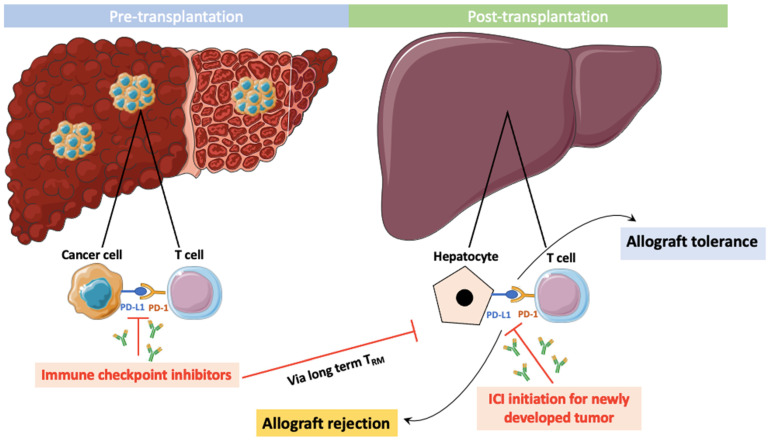
Schematics of pre- and post-transplantation ICI therapy.

**Table 1 cancers-15-04574-t001:** Implicated immune cells in HCC, ICIs, and transplantation.

Cell Type	Implication		
	HCC	ICI	Transplantation
Regulatory T cells (Treg)	- Promote tumor cell evasion  - Function and expansion of: - T cells (CD4, CD8)- B cells- NK cells- APCs- FoxP3 + CTLA-4 + CD4 + enriched in viral-related HCC correlate with poor prognoses	- Treg depletion  - Suppressive activity on PD-1/PD-L1 blockade	- Participate in allograft tolerance
Resident memory T (T_RM_) cells	- Participate in TME homeostasis 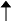 - Expression of ICOS, PD-1, TIGIT, and Tim-3- Association of T_RM_ cell presence in the TME and better oncological outcomes- T_RM_ cell dysfunction participates in tumor development	- High level of CD103 + T_RM_ cells in the tumor correlates with responses to ICIs and prognoses- High expression of ICs in CD8 subgroup	- Nonexhausted T_RM_ cells participate in chronic rejection- Suspected to be responsible for acute rejection after ICI therapy (long-term memory)
Exhausted T (T_EX_) cells	- High fraction of CD8 + PD-1 + expressing CTLA-4, CD39, and LAG-3 correlate with poor overall survival- High fraction in advanced HCC	- Their presence is correlated with poor responses to ICIs	- Suspected to become exhausted after chronic exposition to alloantigens


 = decreased, 
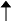
 = increased.

**Table 2 cancers-15-04574-t002:** ICIs before liver transplantation.

Study	Study Type	Number of Patients Receiving ICIs Pre-Transplantation (Rejections)	Age/Sex	Underlying Liver Disease	ICI	Duration	Washout Period (Days)	Rejection Proved by Biopsy	Retransplantation	Postoperative Follow-Up
Schwacha-Eipper, 2020 [22]	Case report	1 (0)	62/M	ALD	Nivolumab	34 cycles	105	No rejection	-	12 months
Nordness, 2019 [19]	Case report	1 (1)	65/M	HCV	Nivolumab	24 months	8	POD 6	No, deceased at POD 10	-
Chen G, 2021 [20]	Case report	1 (1)	39/M	HBV	Toripalimab	10 months	93	POD 2	No, deceased at POD 3	-
Dehghan, 2021 [23]	Case report	1 (1)	65/F	HCV	Nivolumab	15 months	35	POD 10	Yes, POD 34	18
Aby, 2021 [72]	Case report	1 (1)	64/M	HCV	Nivolumab	23 months	16	POD 9	No, high-dose corticosteroids	16 months
Sogbe, 2021 [73]	Case report	1 (0)	61/M	HBV	Durvalumab	18 months	92	No rejection	-	24 months
Tabrizian, 2021 [75]	Case series	9 (2 *)	N/D	HBV	Nivolumab	N/D	1–253	N/D	N/D	N/D
Qiao Z, 2021 [21]	Case series	7 (1)	Mean age 53 +/− 12.1/M	N/D	Pembrolizumab or camrelizumab	N/D	40 on average	POD 11	No, corticosteroids	N/D
Schnickel, 2022 [76]	Case series	5 (2)	60/F 65/M	HCV HCV	Nivolumab Nivolumab	18 months 8 months	35 10	POD 14 <POD 14	No, corticosteroids No, rATG, rituximab, or IVIGs	38 months 3 months
Dave, 2022 [77]	Case series	5 (2)	Mean age 61 +/−6.52/N/D	N/D N/D	Nivolumab Nivolumab	N/D N/D	<90 days <90 days	Yes Yes	Yes, successful No, death 2 months after transplantation	N/D N/D
Kang, 2022 [74]	Case report	1 (0)	14	None	Pembrolizumab	3	138	No	No	96 months
Chen, 2021 [79]	Case series	5 (0)	Mean age 53.2 +/− 5.4/4M, 1F	N/D	Nivolumab	N/D	63.80 ± 18.3	No	No	12 months
Wang, 2023 [78]	Case series	16 (9)	37–67/14 M–2 F	14 HBV 2 ALD	2 nivolumab, 7 pembrolizumab, 4 sintilimab, 2 camrelizumab, and 1 multiple	1–27 cycles	7–184			352.5 (median)

ALD: alcohol-associated liver disease, IC: immune checkpoint, ICI: immune checkpoint inhibitor, N/D: not disclosed, M: male, F: female, POD: postoperative day, rATG: rabbit antithymocyte globulin, IVIGs: intravenous immune globulins, HCV: hepatitis C virus, HBV: hepatitis B virus, and HCC: hepatocellular carcinoma. * One attributed to low immunosuppression levels.

**Table 3 cancers-15-04574-t003:** ICIs after liver transplantation.

Author	Year of Publication	*n*	Acute Graft Rejection Rate	Death Due to Rejection	OS (Months)	Most Commonly Used ICIs	IS While on ICI	Tumor PD-L1 Staining	Indication	Time from Transplant to ICI Initiation (Years)
De Toni [92]	2017	1	No	No	7	Nivolumab	Tacrolimus	/	HCC recurrence	11
Brumfiel [93]	2021	1	No	No	15	Nivolumab	MMF + prednisone + tacrolimus	/	Cutaneous SCC	>21
Bittner [94]	2021	1	Yes	No	>14	Nivolumab	MMF relayed by tacrolimus and everolimus due to rejection	Positive	PTLD	11
Ben Khaled [91]	2021	1	No	No (POD)	/	Atezolizumab/bevacizumab	-	/	HCC recurrence	4
Kondo [95]	2022	1	No	No (POD)	/	Nivolumab	Cyclosporine + MMF	Positive	Hypopharyngeal SCC	>3
Tsung [96]	2021	2	No	No	/	Cemiplimab	Tacrolimus	/	Cutaneous SCC	/
Owoyemi [97]	2020	8	1/4	No (POD)	/	Nivolumab 75% Pembrolizumab 25%	Calcineurin inhibitors alone 65%, tacrolimus + prednisone 13%, MMF and pred 13%, other	/	1/8 SSC, 5/8 HCC, 2/8 melanoma	3
Al Jarroudi [98]	2020	3	No	No	>4 months	Nivolumab	Tacrolimus	/	HCC recurrence	1 to 3
Braun [99]	2020	1	Accelerated chronic rejection	Yes	2	Nivolumab	Tacrolimus	/	Lung NSCLC	3
Anugwom [100]	2020	1	Hepatitis linked to ICIs	No	2	Nivolumab	Tacrolimus	Negative	Metastatic HCC + NSCLC	1
Pandey [101]	2020	1	No	No	>27	Ipilimumab	Tacrolimus	/	HCC recurrence	7.5
Amjad [102]	2020	1	No	No	>24	Nivolumab + prednisone	Tacrolimus + MMF	Positive	HCC recurrence	2
Zhuang [103]	2020	1	No	No	20	Nivolumab	Tacrolimus	/	HCC recurrence	2
Lee [104]	2019	1	Yes	Yes, delayed	/	Nivolumab	Everolimus	/	SCC	1
Chen [105]	2019	1	No	No	/	Pembrolizumab + prednisone	Tacrolimus	/	Metastatic CRC	4
Deleon [107]	2018	5	1/5	/	/	Nivolumab	Sirolimus or tacrolimus or MMF + sirolimus	Positive 1/5	HCC	3.92 (mean)
		2	1/2	/		Pembrolizumab	Sirolimus or tacrolimus or MMF + sirolimus	Positive 1/2	Melanoma	4.3 (mean)
Tio [108]	2018	1	Yes	Yes	/	Pembrolizumab	Cyclosporine	/	Melanoma	/
Nasr [109]	2017	1	No	No	>12	Pembrolizumab	Tacrolimus + MMF	/	HCC recurrence	4
Guoying [110]	2016	1	Hepatitis linked to ICIs	No	/	Pembrolizumab	Tacrolimus + sirolimus	/	HCC recurrence	1
Gassmann [111]	2018	1	Yes	Yes	/	Nivolumab	MMF + everolimus	/	HCC recurrence	2
Rammohan [112]	2018	1	No	No	> 10	Pembrolizumab	Rapamycine + tacrolimus	/	HCC recurrence	3
Kuo [113]	2018	1	No	No	/	Ipilimumab, followed with pembrolizumab	Sirolimus	/	Melanoma	1
Biondani [114]	2018	1	No	No (POD)	/	Nivolumab + prednisone	Tacrolimus + everolimus	/	Lung NSCLC	13
Varkaris [115]	2017	1	No	No (POD)	/	Pembrolizumab	Tacrolimus	/	HCC	8
Friend [116]	2017	2	Yes	Yes	/	Nivolumab	Sirolimus or tacrolimus	Positive	HCC	3 and 4
Dueland [117]	2017	1	Yes	Yes	/	Ipilimumab	Prednisolone	/	Ocular melanoma	1.5
Schvartsman [118]	2017	1	Hepatitis linked to ICIs	No	>6	Pembrolizumab	MMF	/	Melanoma	>20
Morales [119]	2015	1	No	No	>4	Ipilimumab	Tacrolimus	/	Melanoma	8
Ranganath [120]	2015	1	No	No	>5	Ipilimumab	Tacrolimus	/	Melanoma	8
Abdel-Wahab [106]	2019	11	4/11	1/11	/	Ipilimumab/nivolumab/pembrolizumab	/	/	6/11 melanoma 4/11 HCC recurrence	6.87 (mean)

POD: progression of disease, ICI: immune checkpoint inhibitor, SCC: squamous cell carcinoma, HCC: hepatocellular carcinoma, NSCLC: non-small-cell lung cancer, CRC: colorectal cancer.

**Table 4 cancers-15-04574-t004:** Ongoing clinical trials on ICI use in neo/adjuvant setting.

Name		Number of the Study	Study Start	Phase	Main Outcome	Expected Study Termination	Location
Atezolizumab and Bevacizumab Pre-Liver Transplantation for Patients with Hepatocellular Carcinoma Beyond Milan Criteria		NCT05185505	30.01.23	4	Proportion of patients receiving liver transplant experiencing acute rejection	31.10.27	Houston, USA
Atezolizumab and Bevacizumab before surgery for the treatment of resectable liver cancer		NCT04721132	10.02.21	2	Pathologic complete response rate	31.12.27	Houston, USA
Neoadjuvant combination of atezolizumab/bevacizumab versus Neoadjuvant radiation therapy	ADVANCE HCC	NCT05137899	18.10.22	2	Proportion of patients who undergo hepatectomy in each arm	30.06.26	Canada
A study of atezolizumab plus bevacizumab versus active surveillance as adjuvant therapy in patients with hepatocellular carcinoma at high risk of recurrence after surgical resection or ablation	IMbrave050	NCT04102098	31.12.19	3	Recurrence-free survival	16.07.27	International (USA, Canada, Australia, New Zealand, Austria, Belgium, France, Spain, Germany, Italy, Czechia, Netherlands, Poland, Turkey, Peru, Brazil, Costa Rica, Mexico, China, Hong Kong, Japan, Korea, Thailand, Taiwan, Singapore, Russia)
A study evaluating the efficacy and safety of neoadjuvant immunotherapy combinations in patients with surgically resectable hepatocellular carcinoma	MORPHEUS-NEO HCC	NCT05908786	01.09.23	Ib/2	Major pathologic response rate	31.03.25	USA, Canada

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
