# Peer review of "Immunotherapy and Liver Transplantation: A Narrative Review of Basic and Clinical Data"

_cancers, 2023, doi:10.3390/cancers15184574_

Round 1
Reviewer 1 Report
This manuscript gives an excellent overview on the pre-clinical and clinical data on the use of immune checkpoint inhibitors in HCC prior and after liver transplantation.
The manuscript is well and clearly and written. It includes all available references on this topic, therefore is very useful for the reader.
I would like to congratulate the authors for this manuscript.
Author Response
Thank you very much for your report.
Reviewer 2 Report
This paper is an interesting review of the multiple therapeutic strategies available to adequately treat patients diagnosed with HCC. In addition, the authors expose the importance of immune checkpoint inhibitors (ICI) in cases of advanced stage of HCC in order to perform a liver transplant. Based on this review, prospectives studies are necessary to analyze the efficacy of ICI in HCC associated with liver transplantation .
Author Response
Thank you very much for your review and report. We totally aggree on the need of prospective studies and are waiting for the results of the ongoing trials.
Reviewer 3 Report
I appreciated this work very much and I think it is quite clear although the subject is complicated. The treatment of advanced HCC is very fascinating and still more the possibility to down-stage it, leading it to transplant. Immunotherapy introduces new challenges for the future and I think that such a kind of studies, more and more deep, have to be promoted.
Many compliments also for the figures.
I think that minor editing of English language is enough to accept this work in the present form
Author Response
Thank you for your report and remarks. We used the editing program of MDPI and submitted the manuscript after english edited.
Reviewer 4 Report
The field of HCC management is undergoing rapid change, with the most recent advances linked to the introduction of immunotherapy as both primary and adjuvant lines of treatment
This review is clear to show the advantage and side effects od the ICI use and call the attention to early liver rejection when the liver transplantation was carried out after no long ICI administration.
none
Author Response

(The authors gave the same response as above.)
